# Fitness Cost of Antiretroviral Drug Resistance Mutations on the *pol* Gene during Analytical Antiretroviral Treatment Interruption among Individuals Experiencing Virological Failure

**DOI:** 10.3390/pathogens10111425

**Published:** 2021-11-03

**Authors:** James R. Hunter, Domingos E. Matos dos Santos, Patricia Munerato, Luiz Mario Janini, Adauto Castelo, Maria Cecilia Sucupira, Hong-Ha M. Truong, Ricardo Sobhie Diaz

**Affiliations:** 1Department of Medicine, Federal University of São Paulo, São Paulo 04039-032, Brazil; jhunter@unifesp.br (J.R.H.); domingos.matos@yahoo.com.br (D.E.M.d.S.); Patricia.Munerato@cepheid.com (P.M.); Janini@Unifesp.br (L.M.J.); adauto.castelo@unifesp.br (A.C.); cecilia.araripe@unifesp.br (M.C.S.); 2Department of Medicine, University of California, San Francisco, CA 94158, USA; Hong-Ha.Truong@ucsf.edu

**Keywords:** antiretroviral virologic failure, antiretroviral resistance, analytical treatment interruption, fitness cost

## Abstract

HIV cure studies require patients to enter an analytical treatment interruption (ATI). Here, we describe previously unanalyzed data that sheds light on ATI dynamics in PLHIV (People Living with HIV). We present drug resistance mutation dynamics on the pol gene among individuals with antiretroviral virological failure who underwent ATI. The study involved a 12-week interruption in antiretroviral therapy (ART), monitoring of viral load, CD4+/CD8+ T cell counts, and sequencing of the pol gene from 38 individuals experiencing virological failure and harboring 3-class resistant HIV strains: nucleoside reverse transcriptase inhibitors (NRTI) non-nucleoside inhibitors (NNRTI), and protease inhibitors (PI). Protease and reverse transcriptase regions of the *pol* gene were sequenced at baseline before ATI and every four weeks thereafter from PBMCs and at baseline and after 12 weeks from plasma HIV RNA using population-based Sanger sequencing. Average viral load increased 0.559 log_10_ copies per milliliter. CD4+ T cell count decreased as soon as ART was withdrawn, an average loss of 99.0 cells/mL. Forty-three percent of the mutations associated with antiretroviral resistance in PBMCs disappeared and fifty-seven percent of the mutations in plasma reverted to wild type, which was less than the 100% reversion expected. In PBMC, the PI mutations reverted more slowly than reverse transcriptase mutations. The patients were projected to need an average of 33.7 weeks for PI to revert compared with 20.9 weeks for NRTI and 19.8 weeks for NNRTI. Mutations in the pol gene can cause virological failure and difficulty in re-establishing effective virological suppression.

## 1. Introduction

In the late 1990s and early years of this century, researchers studied the clinical benefit that analytical treatment interruption (ATI) might offer HIV-1 patients experiencing antiretroviral virological failure given the limited options of salvage therapy available at that time. However, these studies led to the conclusion that complete ATI cannot provide sufficient clinical benefit to justify its use as an adjunct to antiretroviral therapy (ART) programs [1,2,3,4,5].

At the time the data were gathered for this study, researchers believed that ATI would hasten the reappearance of wild type virus, which in turn would facilitate the subsequent reintroduction of ART [6,7,8,9]. ATI was soon considered an ineffective treatment strategy due to the HIV persistence in long-lived latent cells. This data set has only been sporadically studied since then [10,11].

### 1.1. Origin of Study

From June 1999 to June 2000, the authors collected samples from 38 HIV-1 treatment-experienced patients experiencing antiretroviral virological failure who agreed to participate in a study of ATI. This paper analyzes the original data in a more consistent way than conducted previously. Although we no longer consider ATI a viable treatment strategy, there is still much to be learned from these data in terms of the evolution of resistance patterns in the reverse transcriptase and protease portions of the HIV-1 genome. The difference between the virus genotype with and without the selective pressure of ART offers important information about effective treatments for the future. 

Once infected, the patient’s HIV-1 begins to evolve and continues to do so even as it faces long-term replication suppression [12,13]. Given that ART suppresses the susceptible HIV strains, mutational variants appear that have greater replication fitness. These variants permit the viral load of the patient to increase, which in turn leads to greater inflammation and transmission opportunity [14,15,16,17,18,19,20,21]. 

Some anatomical sites such as the central nervous system and genital tract may be inaccessible to some classes of antiretroviral drugs. For example, the brain absorbs protease inhibitor (PI) medications poorly [22]. These inaccessible compartments, along with resting CD4+ lymphocytes, create HIV-1 resistance havens and permit movement of virus between compartments with measurable resistance levels [23,24].

Viral fitness, the ability to create more adapted and highly replicating infectious progeny, is inversely related to the number of incorporated antiretroviral resistance related mutations [25,26,27]. In the absence of drugs, the wild-type virus or a less resistant one has more capacity to outgrow the resistant strains [28,29,30].

### 1.2. Objectives

The current study describes viral load and CD4+/CD8+ T cell count dynamics. We also characterize the drug-resistant mutation dynamics in the plasma and PBMC. We focus on mutations associated with HIV-1 resistance at the reverse transcriptase and protease regions of the pol gene in DNA and plasma RNA. The expectation of the study team was that, since the use of ART repressed the wild-type virus, it would proliferate on withdrawal of ART and that the resistant strains would disappear due to the greater replicative fitness of the wild type. 

Finally, we report on the presence of cell activation markers HLA-DR and CD38 on CD8+ T cells. These proteins are associated with activation of latent infected CD4+ T cells, making them available for attack by the patient’s immune system. Prior research has indicated that expression of these activation markers is associated with an increase in viral load and a decrease in CD4+ T cell counts [31,32,33,34,35,36]. These studies, however, show the paradoxical effect of ART on cell activation. ART suppresses the viral load, which allows the CD4+ T cells to recover, which in turn creates more targets for the virus to attack. Since the wild-type virus that the ART is suppressing is now at generally undetectable levels, the virus strains that can infect new CD4+ T cells are generally resistant to the therapy and thus are better able to proliferate. Further complicating the clinical picture for patients, activating latent infected cells is a key challenge for researchers and clinicians. Without activation, these cells would remain latent and invisible to ART drugs. A team of researchers has called this phenomenon the “paradoxical Pas de Deux” [34].

Previous use of antiretrovirals before the experimental period revealed a variety of antiretrovirals available then. Appendix A shows how many patients used each antiretroviral.

## 2. Results

### 2.1. Patients

This study included 36 patients with HIV-1 3-class antiretroviral resistance mutations and 2 patients with chronic HIV-1 infection but no detectable HIV-1 resistance related mutations. These latter patients served as controls. There were one male and one female control subjects. The mean age of the controls was 37.0 years (standard deviation of 1.41 years). The patients harboring resistance included 28 males (77.8%) and 8 females (22.2%). The mean age of the resistant group was 38.75 years (standard deviation of 7.77 years). All patients had a virus of subtype B except for patients 3 and 53, who had combination subtype B/F.

### 2.2. Reversion Categories

For the reverse transcriptase and protease regions of the pol gene, the patients were divided based on the degree to which they returned to the wild type. These categories are high reversion, low reversion, and non-reversion. We established the reversion group assignment on the return to wild-type pattern in the reverse transcriptase region of the plasma sequences. The decision rule for assignment to each group was based on a comparison of the number of mutations that the patient experienced between baseline (week 0) and week 12. The “None” group had an equal or greater number of mutations at the end than at the baseline. The “Low” group had fewer but not zero mutations. In other words, they had at least one mutation but less than they had at baseline. The “High” group experienced zero mutations at week 12 after showing one or more at baseline. In this scheme, the controls were those with zero mutations at baseline.

The last line of Table 1 shows the count of patients classified in each reversion group, and Figure 1 shows the average number of mutations per patient for each of the groups within the four classes of blood fraction (PBMC or plasma) and region on the pol gene (reverse transcriptase or protease). The “All Resistant Subjects” column includes all three reversion groups.

### 2.3. Viral Load Dynamics

Table 1 also shows the evolution of viral load (measured by a log_10_ transformation of the raw data) over the twelve weeks of the study for all four groups (three reversion groups and the control group). The data underlying these results, the viral loads, CD4+ counts, and CD8+ counts for all patients at all sample collection times are included in the Appendix A. The means and standard deviations by groups are provided in Appendix A.

Figure 2 shows the mean profile of the viral load dynamics for all of the reversion groups. The mean viral load increased during the test period for the high and low groups by more than 0.5 log_10_. However, the viral load of the no reversion group rose until week 8 and then declined. It only increased over the study period by 0.136 log_10_. Overall, the groups, including the controls, increased by an average of 0.568 log_10_. In addition, the high and low reversion groups’ viral load increase was statistically significant. A paired two-sample t-test of the difference between viral load in all resistant patients (groups High, Low, and No taken together) at baseline and at week 12 had a value of t = 6.2919 with 35 degrees of freedom and *p* < 1.0 × 10^−6^. The linear increase in viral load in the current cohort contrasts with the rapid rise in viral load reported in a previous study [9].

### 2.4. CD4+ and CD8+ T Cell Dynamics

Figure 3 shows the effect of treatment interruption on CD4+ T cell counts of patients in the three reversion and control groups. In this case, patients showed an immediate and consistent decline in cell counts over the 12 weeks of the experiment. The decrease was an average of 99.0 cells/mL among patients showing drug-resistant mutations at baseline. By the end of week 12, the three reversion groups showed mean CD4+ levels below 200 cells/mL.

Focusing on the three groups that showed initial resistance mutations, the decline in CD4 levels between baseline and week 12 was statically significant (Wilcoxon signed-rank test with continuity correction, *p*-value < 1.0 × 10^−6^.

The CD8+ T cell results roughly mirrored those of CD4+ cells for the low reversion and control groups. The no reversion and high reversion groups showed slight overall declines, with the no reversion group showing a slight increase between weeks 4 and 8. Overall, the resistant groups experienced a mean decrease in CD8+ levels of 145.4 cells per mL with a standard deviation of 409.7 cells. The Appendix A contain the figure showing the CD8+ results (Appendix A).

### 2.5. Reversion to Wild Type among Resistant Patients

The number of mutations declined during the ATI, as expected for the reversion groups in both plasma and PBMC blood fractions and in both the reverse transcriptase and protease genomic regions. However, only in reverse transcriptase mutations in plasma (high reversion group) did the number of mutations go to zero. In all other cases, there remained a small number of mutations per patient in week 12.

That the resistant mutations would revert to the wild-type virus due to ATI among individuals with antiretroviral failure was only partially supported by the data from this study where the proviral DNA profile is concerned. We limited all of the analyses of reversion to wild type to the 36 patients who showed resistant mutations. However, we include in the Appendix A, the mutations pre- and post-ATI for all of the patients as well as the antiretroviral drugs (including the different regimens) that they used before the ATI. This table also specifies the subtypes of HIV for each patient.

### 2.6. Reversion to Wild Type in Plasma

The plasma sequences collected at baseline and study end (week 12) show an uneven reversion pattern despite the expectation that all mutated codons would revert to wild type after 12 weeks of ATI. Among the patients showing drug resistance, a mean of 14.95 codons evolved between the start and end of the treatment interruption (95% confidence interval of 10.61 to 19.28). Of the 434 amino acids in the sequence studied, 111 different codons evolved during the study. Some of these 111 codons had variant amino acids at the end of the survey, so the number of different mutations during the treatment interruption totaled 164. For example, codon 20 showed an evolution of R10K in five patients and I10K in two other patients. 

We focused our analysis on those codons that evolved in more than one patient. Table 2 lists these codons. We have not included codon changes that came from only one patient, as they may be as much a result of a sampling or measurement anomaly as they are of an underlying biological process. The format for referring to the amino acids shows the mutated state (pre-ATI) first and then the evolved state, which is the wild-type amino acid for that location in all cases. For example, the listing for codon at position pr10 is generally referred to as L10I, with the wild-type amino acid coming before the location number and the mutation after. The “pr” designation refers to protease inhibitor resistant codons and “tr” refers to reverse transcriptase resistant codons.

Although our sample was drawn well before IAS began to report drug resistance mutations, we tested whether mutations in patients with drug resistance are consistent with the 2017 IAS list of drug-resistant mutations. In 35 of the 40 codons in the table (87.5%), the patients reverted to wild type as defined in the IAS 2017 interpretations.

### 2.7. Protease Inhibitor (PI) Mutations at DNA Level

We conducted a parallel analysis of the PBMC samples from the 36 resistant patients for the evolution of both protease inhibitor and reverse transcriptase inhibitor resistance mutations. We hypothesized that the high reversion level that we would see at the plasma level would not be observed in this compartment. 

Of the PI codons on the IAS list, we initially had data on 22. These were codons 10, 20, 24, 30, 32, 36, 46, 47, 48, 50, 53, 54, 63, 71, 73, 77, 82, 84, 85, 88, 89, 90, and 93. We eliminated codons 32, 47, and 50 from our analysis as they either had 0 or 1 tests that showed that mutation and therefore did not allow comparison. For codon 89, only patient 3 exhibited a mutation. Codons 77 and 88 showed an increase in the number of patients exhibiting the mutation during the 12 weeks and thus had positive slopes in the regression. Accordingly, it was not possible to calculate a time to 0 patients using regression. However, the Kaplan–Meier survival analysis result showed a probability of the survival of the mutation at codon 77 of 0.318 (95% CI of 0.173–0.587) and at codon 88 of 0.176 (95% CI of 0.063–0.493). Given the overall sample size of 36 and the variable number of patients who showed the mutation (between 7 and 11 for codon 77, and between 1 and 4 for codon 88), it is not possible to tell if this result is due to a sampling error or to some fundamental observation about these codons. 

We evaluated the consistency of results between the time to 0 mutations for each codon and the survival analysis. We found a correlation between these values of 0.69. This correlation was significant (*p* = 0.00013) using a Pearson’s Product Moment Correlation test. We included a figure with this result in the Appendix A. 

Table 3 shows the number of weeks it would take for each mutated codon to return to wild type at the DNA level via a regression line projection for all three types of drug resistance. Rather than being a specific forecast of how long the mutations would persist in the absence of ART, the projection indicates the mutation’s general tendency. The strength of the regression model for each codon is expressed in the R^2^ column. This measure indicates the percentage of overall variance that the model expresses. The table also shows the probability that the mutation would survive after the 12-week interruption derived from the Kaplan–Meier analysis. In other words, it estimates the likelihood that the mutation would not revert to wild type. As seen in the table, those codons with smaller mutation survival probabilities also had generally shorter projected periods to 0 patients carrying the mutation.

The mean number of weeks for the protease inhibitor mutations is 34.5 weeks with a standard deviation of 27.3 weeks. Figure 4 illustrates the transition to wild type virus for codons 46 and 90. These were chosen for display because they have the highest coefficients of determination (R^2^) among the PI mutation codons. Graphs for the other codons are found in the data repository for the study (https://github.com/jameshunterbr/ATI, accessed on 5 October 2021).

This figure shows the number of patients with a mutation at each test and a regression of these points. The regression equation shows the parameters determining the line, and the R^2^ indicates the degree to which the line explains the points.

### 2.8. Nucleoside Reverse Transcriptase Inhibitor (NRTI) Mutations at DNA Level

We had data on nine of the IAS’s list of 15 NRTI codons (41, 65, 67, 70, 75, 184, 210, 215, and 219). We eliminated codon 65 from further analysis as it had only one patient with a mutation at this location at week 0. Codon 75 just had two data points: one in week 0 and one at week 12. Table 4 shows the trend toward reversion to wild type for the NRTI codons for the remaining seven codons and the probability that the mutation would survive after the 12-week ATI. The overall mean for reversion time was 20.9 weeks with a standard deviation of 4.8 weeks.

Codon 184 (M184V) comes first in the Stanford University database of NRTI mutations because of its 100-fold reduction in patient susceptibility to Lamivudine/Emtricitabine but its enhancement of the susceptibility to Zidovudine and Tenofovir Disoproxil Fumarate [37]. The other six NRTI mutations in our study are all thymidine analog mutations (TAMs). Figure 5 shows the number of weeks that we project it would take to completely extinguish the M184V mutation and the two groups of TAM mutations (TAM 1 including the mutations at codons 41, 210, and 215 and TAM 2 including codons 67, 70, and 219). The R^2^ measure indicates the relative variability within the group.

Both TAM groups have mean values very close to each other. According to a sample of 3 per group, there is no difference between the two TAM types with a Wilcoxon Rank Sum test (*p* = 1.0). However, the TAMs considered together take a significantly longer time to revert to wild type than codon 184 according to a one-sided Wilcoxon Signed Rank test (*p* = 0.016). This started with 28 patients exhibiting the Valine mutation, the loss of one carbon atom that distinguishes it from the Isoleucine wild type. By the end of the 12 weeks, the number showing this mutation had dropped to 8, and our linear regression model for this codon projected it would be eliminated by week 16. The TAM groups showed fewer patients with the mutations at the start of the interruption (20.7 for TAM 1 and 17.7 for TAM 2), declining to 8.3 and 7.7, respectively, at the end of the study. The higher rate of reversion for I184V than the TAMS can be seen in greater detail in the panels of Figure 5, which show all of the NRTI codons.

### 2.9. Non-Nucleoside Reverse Transcriptase Inhibitor (NNRTI) Mutations at DNA Level

Our data included 7 of the 16 non-nucleoside reverse transcriptase inhibitors (NNRTI) on the IAS list (103, 106, 108, 179, 181, 188, and 190). We eliminated codons 106 and 108 from the analysis as they had insufficient data to permit a regression to be performed. We also removed codons 179 and 188 from the analysis as the counts of patients with the mutation only varied between 1 and 3, which strongly suggested that our sample was not typical of the population. The last line of Table 5 shows the number of weeks required for the three remaining NNRTI mutations to revert to wild type.

The average number of weeks for NNRTI mutations to revert to wild type was 19.8 weeks with a standard deviation of 2.2 weeks. Figure 6 shows the rate of reversion for the NNRTI mutations. 

Although the graphs and tables we display here show the range of rates of reversion to wild type among the codons studied, the average for each of the three classes of antiretroviral medications indicates that this is a prolonged process that may last as long as 8 months (in the case of protease inhibitor mutations). Table 4 shows these averages.

### 2.10. CD38 and HLA-DR T Cell Activation Markers

The study sample showed a greater presence of CD38+ and HLA-DR+ proteins on CD8+ T cells in patients who returned to wild type. The no reversion group, whose mutations remained constant despite the treatment interruption, expressed a significantly lower level of these cell activation biomarkers (measured by the percentage of CD8+ T-cells with both CD38 and HLA-DR present) than did the high reversion group. Table 5 and Figure 7 show these summary measures. 

This result parallels the observation that cell activation markers would be expressed in greater quantity in untreated HIV-1 patients than in patients receiving treatment [32]. 

To create an index of cell activity absent viral load, we corrected the cell activation marker levels for the log_10_ viral load by dividing the cell activation markers’ expression by the log_10_ viral load. As Table 5 and the second panel of Figure 7 show, there is still a difference among the groups, but it is small and not statistically significant.

Table 6 uses the mean of all four visits to determine the cell activation level. If we look at the evolution of the cell activation from baseline to week 12 in the no reversion and high reversion groups (where the difference was significant above), we find that there is an apparent difference between the two groups, but it is not significant when calculated for either the raw values or the corrected cell activation values, as seen in Table 6. While the No Reversion group shows more significant growth in cell activation between baseline and the ATI end, the difference is not significant.

We compared the T cell activation level with the viral load and CD4+ T-cell levels for all the patients. Overall, the correlation with viral load was low but statistically significant (Spearman’s rank correlation, r = 0.46, *p*-value < 1.0 × 10^−6^). However, the correlation between cell activation and CD4+ T-cell levels is stronger and statistically significant (Spearman’s rank correlation, r = −0.52, *p*-value < 1.0 × 10^−6^) showing that the higher the T cell activation, the lower the CD4+ T-cell counts. Figure 8 shows these correlations.

Previous research has found a positive association between viral load and CD38+ CD8+ T cells [31]. Our correlation results do not show the same level of significance in such an association. 

However, our data regarding cell activation and CD4+ T cell counts closely parallel previous studies analyzing cell activation in subpopulations of HIV patients [31,38]. Our data mirror those of treatment-naive patients in these studies. In the current study, patients at baseline are under treatment, and at the end of the study, they should be similar to untreated patients in these other studies.

## 3. Discussion

As was expected, the viral load increased for patients subject to ATI, and CD4+ T-cell counts decreased. The viral load increase results from an increase in active viral replication after ATI. This is an important factor in replenishing the proviral reservoir even for this short period. This has been shown previously in this same patient population [11]. One factor that is featured in other studies of the impact of ATI on viral load is a positive relationship between the rapidity of the viral rebound and the size of the expressed, rather than latent, HIV reservoir in patients [39]. We cannot comment on that result as we lacked the data. However, it should be considered in future studies. Of note, the two patients in the control group also experienced a biologically significant increase in the viral load (greater than 0.5 log_10_) without changing their HIV genetic profile, which argues in favor of the activity of antiretrovirals in use, although probably without a good adherence to current treatment. 

### 3.1. Process of Reversion to Wild Type

The picture for reversion to wild type is not clear. This study shows that reversion is not as direct a process as previously thought for resistance-related mutations selected by all three classes of antiretrovirals studied [7,30]. While the drug-resistant strains did show movement toward the wild type, only a sub-group achieved complete reversion and then only in plasma (high reversion group). At the end of the study, patients still had several resistant-related codons that differed from the wild type in the PBMC fraction. Third, the projected time needed for drug-resistant strains to revert to wild type was not consistent across the mutations and was proven to be an extended period for several mutations at the DNA level. We encountered cases where mutations became more common among the patients during the interruption rather than rarer at the DNA level. 

What enables such a result? First, the period chosen for the treatment interruption—twelve weeks—may not be sufficiently long for complete reversion to the wild-type profile at the DNA level, although it was the standard for ATI studies at the time. A 1992 study from Sweden indicates that patients taking zidovudine (AZT) who developed resistance to the medication and subsequently abandoned treatment still carried some of the mutations as long as nine months after stopping treatment [40]. As noted above, the mean times until the entire cohort reverted to wild type was projected to range from 19 to 33 weeks depending on the class of mutation being modeled. While NRTI mutations are congruent with the Swedish study, two codons selected by PI or be considered polymorphisms (36 and 63) had complete reversion projections exceeding 1.8 years. These two polymorphisms also showed the highest probabilities of the survival of the mutation among the codons studied (0.483 and 0.647). 

We also need to consider why each mutation disappears with its dynamics. There appears to be little or no uniformity in the rate of reversion. Theoretically, once ART’s selective pressure that gives an advantage to the resistant mutations is withdrawn, the wild-type strain should reemerge rapidly. Interestingly, once ART is stopped and its selective pressure withdrawn, viruses with distinct resistant-related mutation profiles may reemerge as latent cells resume replication and/or based on the differential replicative capacity of distinct strains in each HIV quasispecies. When single genome amplification of antiretroviral resistant strains is performed, mutations that are detected in the population sequence analyses may not be present in all HIV strains but dispersed among distinct viruses of the HIV quasispecies at a given moment [41]. We must also consider that, for experienced patients with virologic failure to multiple HIV regimens, ancient HIV strains with different mutation profiles may exist in individual cells or groups of cells. 

### 3.2. Recombination

After antiretroviral treatment interruption, HIV recombination at the pol gene may dynamically occur in distinct HIV strains with resistant and wild-type profiles, which may in turn seed new proviral populations. As mentioned before, studies exploring partial treatment interruption show that, once NRTIs are withdrawn and PIs are maintained, wild-type HIV emerges at the reverse transcriptase region of the pol gene while the protease region remains resistant [42]. In contrast, when NRTIs are maintained and PIs are interrupted, the virus is not able to recombine, and therefore, resistance mutations stays present at both reverse transcriptase and protease regions even with that absence of PI selective pressure. One explanation for this is that HIV recombines when NRTIs are suspended, and PIs are maintained. The recombination leads to a virus that is wild-type at reverse transcriptase and resistant at protease. Our study may add a new component to this equation since PI mutations lasted longer during ATI in the context of multi-resistant HIV. 

Nonetheless, we have previously demonstrated for this same patient population during the ATI period that the disturbance caused by interrupting the selective pressure imposed by antiretrovirals at the pol gene resulted in ancestral viral progeny activation in a very distinct (and distant) HIV genomic region such as the gp120V3 HIV-1 env region, which repopulated the cell reservoir [43]. 

### 3.3. Persistence of Mutations versus Evolution to Wild Type

In contrast with what has been previously demonstrated in vitro, PI-related mutations’ persistence was longer than the persistence of NNRTI-related mutations. As NNRTI mutations do not impair the HIV-1 fitness as with NRTIs and PIs mutations [44,45], it would be expected that NNRTI mutations would last longer. The fact that NNRTI mutations do not represent a fitness cost for HIV has been implicated in the higher prevalence of transmitted drug resistance related to this class [46]. 

### 3.4. M184V as a Special Case of Mutation

Another feature that deserves further attention is the fact that M184V waned more rapidly than other studied mutations. Although transmitted drug resistance mutations tend to persist in general, M184V tends to revert to wild type in this specific population [47]. It is also conceivable that the APOBEC-related hypermutation plays a role in this process, as valine is encoded by the codon GTG and methionine is encoded by ATG.

We recognize that a more detailed analysis of HIV quasispecies in this study would greatly benefit from single genome amplification or next-generation sequencing (NGS) tools. It has been recently demonstrated using NGS that transmitted drug resistance mutations representing small percentages of the viral population do not persist during infection because they are negatively selected in the first year after HIV-1 seroconversion [48]. In this context, a recent study exploring the extinction dynamics of transmitted drug resistance mutations using next-generation sequencing showed that these mutations are rare at a 20% threshold of detection [49]. Only the M41L (NRTI) and the K103N (NNRTI) mutations appeared in more than 1% of their samples for the resistance class. With a total of 36 resistant patients, our study’s higher prevalence rates, shown in Table 4 above, may reflect the smaller sample size we worked with as well as the greater sensitivity of the NGS techniques used in the other study. 

We previously described that there are two pathways for NNRTI resistance depending upon the drug used. The use of efavirenz primarily selects for the 103 mutation, which is accompanied by the mutations at positions 100 and 225, whereas nevirapine predominantly selects for the 181 mutation, which is generally accompanied by the 101 and 190 mutations [25]. Here, we have shown that mutations at codons 103 and 181 revert at the same timing at the PBMC level. However, the mutation at position 190 reverts faster in this study (mean of 17.5 weeks in contrast with 20.2 weeks for K103N), which is in accordance with the extinction dynamics study referenced above. Compared with other classes, NNRTIs generally have a longer half-life in plasma, averaging 40–55 h for efavirenz [50] and 25–30 h for nevirapine [51]. The host’s genetic profile mainly drives efavirenz’s longer half-life in plasma in the polymorphism of the CYP-2Bc allele. Nonetheless, the longest known half-life of NNRTIs did not affect the reversion time of mutations in PBMCs compared with other drugs from other antiretroviral classes [52].

We also recognize that more intensive cell population evaluations, additional immunologic markers, or direct HIV fitness evaluation could provide additional insights into the mechanisms of viral dynamics after ATI among patients experiencing virological antiretroviral failure. However, the number of patients evaluated here is larger than what has been described in other similar studies, and this is the first study analyzing the impact of such a strategy at a DNA level at multiple time points compared with the plasma compartment, where we sequenced only the baseline and study completion (12-week) time points. We, therefore, have been able to show how the reversion of the virus to a wild type after ATI among individuals failing antiretroviral therapy is not a clear and consistent process across the full range of pol codons studied. We have also provided insight into the persistence of distinct resistant-related mutations selected by different antiretroviral classes. 

### 3.5. Other Areas of the Viral Genome

We studied specific areas of the pol gene related to their known patterns of resistance to the three classes of ART medications in common use at the experiment time. However, our results suggest that ART and subsequent treatment interruption can induce other viral genome changes in which the impact on resistance profiles is not yet well understood. The region between codons 293 and 560 of the reverse transcriptase region is one area of interest. For example, a small number of studies have indicated that Tyr-318 is subject to a range of mutations that show some history of resistance to NNRTIs [44]. Another study suggests that the tat, rev, and nef viral genes evolve during a treatment interruption, although ART drug resistance is not known [53]. 

## 4. Materials and Methods

### 4.1. Patients

Thirty-six HIV-1 treatment experienced patients who were treated at the Federal University of Sao Paulo, Brazil in 1999, agreed to participate in the study on ATI. Two other patients under the same conditions, who suffered from toxic side effects but had no detected HIV-1 resistance mutations, submitted to ATI during the same period and were included in the evaluation. The samples were collected from June 1999 to June 2000. All patients were followed for 12 weeks. The total sample size of 38 was large for studies of this type at the time it was carried out. A non-systematic review of 13 studies on treatment interruption that parallel this study in intent and method published between 1999 and 2004 had a median number of participants of 24 and an interquartile range of 28 (10–38). Of the studies reviewed, only four had more or the same number of participants as the present study. See Appendix A. 

Eligible male and female patients met the following criteria: 18 years or older; experienced multiple ART failures while using nucleoside analog reverse transcriptase inhibitors (NRTIs), non-nucleoside analog reverse transcriptase inhibitors (NNRTIs), and protease inhibitors (PIs); had detectable viral loads above 5000 copies/mL; and showed no signs of AIDS. 

### 4.2. Design

At the time the experiment took place, the design was an interventional study based on the hypothesis that, if the ART drugs were withdrawn, the wild-type virus would replicate and eliminate the presence of resistant strains as the wild type had greater replicative fitness. All patients enrolled on ATI at day zero started a prophylactic regimen against opportunistic agents (data on file). CD4+ T cell counts were taken at 15-day intervals; viral loads were performed monthly; the protease and reverse transcriptase regions of the pol gene were sequenced every four weeks from PBMC; and two plasma genotyping tests were conducted at baseline and week 12 from protease: reverse transcriptase and the p17 region of gag gene. No patient presented AIDS-related symptoms during the study period.

### 4.3. Plasma HIV Viral Load, CD4+ T Count, and Cell Activation Markers on CD4+ and CD8+ T Cells

HIV-1 RNA in plasma was measured using the NASBA methodology (Organon Teknika, Borstel, Holland), according to the manufacturer’s instructions. This assay has a limit of detection of 80 copies/mL.

At each lymphocyte count, we determined sub-populations of CD4+ and CD8+ T cells by three-color flow cytometry. We used fresh samples for direct staining with appropriate monoclonal antibodies (mAb): CD4-PerCP, CD8-PerCP. Mouse immunoglobulin isotypes conjugated with phycoerythrin (PE), or fluorescein isothiocyanate (FITC) were used as negative controls for non-specific binding. The stained cells were analyzed on a FacScan flow cytometer. 

### 4.4. Sequencing of the HIV-1 Pol Gene 

For the analysis of genotypic resistance of PBMC, proviral DNA was extracted from cells using QIAamp DNA Blood Mini kit (QIAGEN Inc., Santa Clarita, CA, USA) and plasma RNA was purified using QIAamp RNA kit (QIAGEN Inc., Santa Clarita, CA, USA) and reverse-transcribed using as previously described [54]. The proviral DNA and plasma cDNA were submitted to an “in house” nested-PCR reaction as previously described for pol fragments [55]. 

The nested-PCR fragments were purified by PCR purification kit (QIAGEN Inc., Chatsworth, CA, USA) and sequenced using Thermo Sequenase fluorescent-labeled primer cycle sequencing kit—RPN 2436 (Amersham Pharmacia Biotech UK Limited, London, UK). The primers were labeled by CY5.5. The products of the sequence reaction were analyzed using a DNA—Long Reader Tower (Visible Genetics Inc.). Base-calling was performed by GeneObjects software (Visible Genetics Inc., Version 3.0, 1998, Toronto, ON, Canada), and the bases were aligned and assembled using GeneLibrarian software (Visible Genetics Inc., Version 3.0, 1998, Toronto, ON, Canada). 

The resulting sequences for each sample were interpreted using the 2017 edition of the IAS–USA drug resistance mutations list [56]. 

Nucleic acid isolation, reverse transcription-PCR amplification, and sequencing of the HIV-1 protease and reverse transcriptase regions of the pol gene were performed using ViroSeq^TM^ HIV-1 Genotyping System v2.0 (Celera Diagnostics, Alameda, CA, USA) per the manufacturer’s instructions. All codons of the HIV-1 protease region were sequenced. The reverse transcriptase region was sequenced up to codon 320. The HIV-1 gag gene amplification products were sequenced in an ABI 3100 Genetic Analyzer (Applied Biosystems, Waltham, MA, USA).

### 4.5. Statistical Analysis

The analysis compared the viral load and CD4+ T cell count dynamics considering patients’ reversion categories. The dynamics of the protease and reverse transcriptase mutations were analyzed by linear regression and Kaplan–Meier survival analysis [57,58,59,60,61]. Given that we had only four time data points (baseline and weeks 4, 8, and 12), we were concerned that regression projections forward of the time of return to wild type would be overly aggressive. While the Kaplan–Meier survival analysis gives an alternate view of the mutations’ survival against their transformation into wild type, it is not intended for projection [62]. 

The data analysis was conducted using the R Statistical Computing System on a Macintosh (version 3.5.0) [63]. Algorithms conducted regressions within the base R system. Survival analyses were performed using the survival package [47]. Sequences and other biological analyses were conducted using R packages from the Bioconductor project [64] and the package seqinr [65]. Given the changes in HIV-1 science in the interval since the data were collected, we chose to concentrate on the mutations that are currently of interest to the HIV-1 research community as determined by the most recent IAS update of drug-resistant mutations [56]. 

We calculated the number of patients who reverted to wild type for each codon in the protease and reverse transcriptase regions on the IAS resistance mutation list for which we had PBMC data. We used the PBMC data for this analysis rather than the plasma data as the former permitted us to examine the evolution over four tests rather than just looking at the beginning and endpoints of the study. 

We had data on 9 of the IAS list of 15 nucleoside reverse transcriptase codons (NRTI) and on 7 of the 16 NNRTI codons on the IAS list. Twenty-two of the thirty-five IAS mutations affecting PI appeared in our sample. We aligned the data that had been gathered in 2000 with the codons on the IAS list. We removed those codons with no patients showing the mutation or had missing values for any of the four visits. 

We then performed a simple linear regression of the number of patients showing the mutation during the study period against the time since ATI initiation and projected from the parameters of the regression line how many weeks would be needed to reach 0 patients, i.e., how many weeks would be required for the mutation to be eliminated from the cohort. In five cases (two for PI mutations, one for NRTI mutations, and two for NNRTI mutations), the regression slope was positive rather than negative, indicating that patients were exhibiting new mutations after the ATI. 

Additionally, we conducted a Kaplan–Meier survival analysis of how long the mutations persisted in the sample and what was the probability that they would persist at the end of the study. 

## 5. Conclusions

These data indicate many interesting features of the mutational patterns of HIV-1 in response to ART selective pressure and its subsequent withdrawal that can be applied to research on understanding fitness features of the HIV vis-a-vis genetic composition of the virus.

## Figures and Tables

**Figure 1 pathogens-10-01425-f001:**
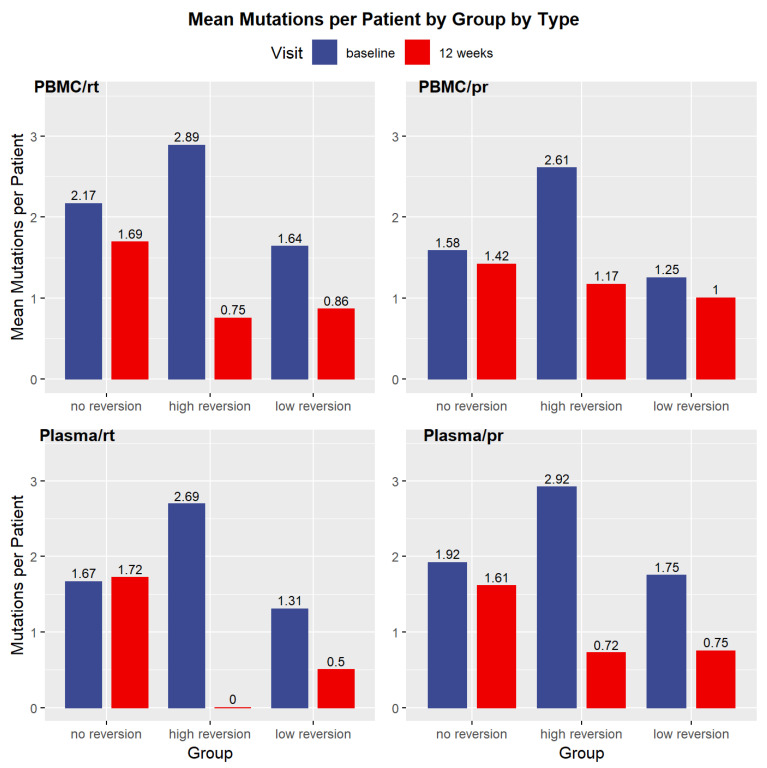
Average number of mutations by reversion group by type. rt = reverse transcriptase; pr = protease. Panels measure the number of mutations present in each blood compartment for the three reversion categories.

**Figure 2 pathogens-10-01425-f002:**
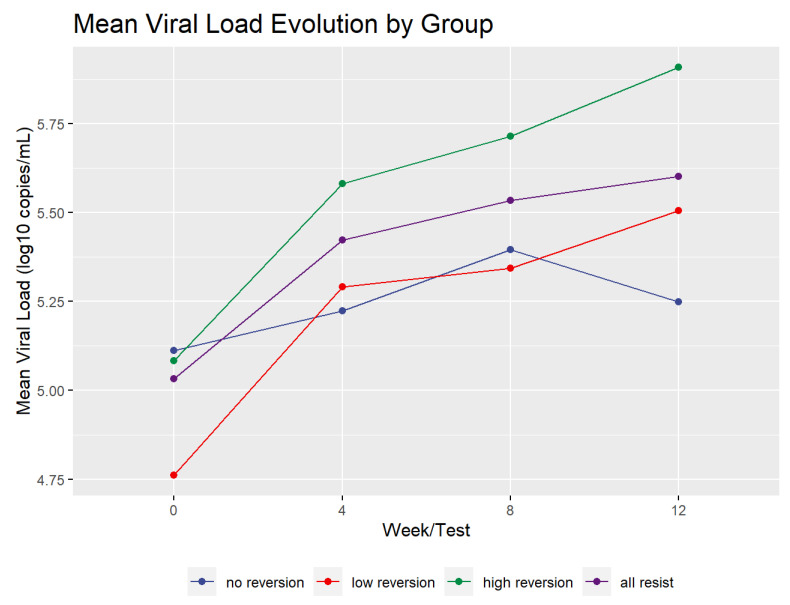
Mean viral load by reversion group.

**Figure 3 pathogens-10-01425-f003:**
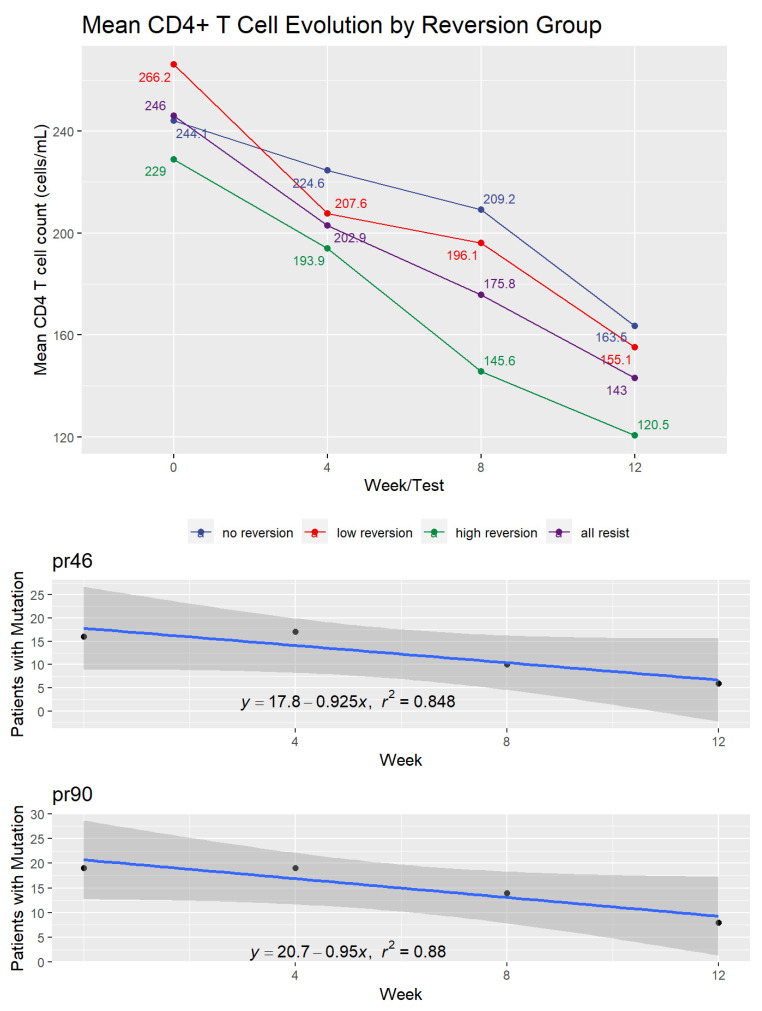
CD4+ T cell evolution by reversion group.

**Figure 4 pathogens-10-01425-f004:**
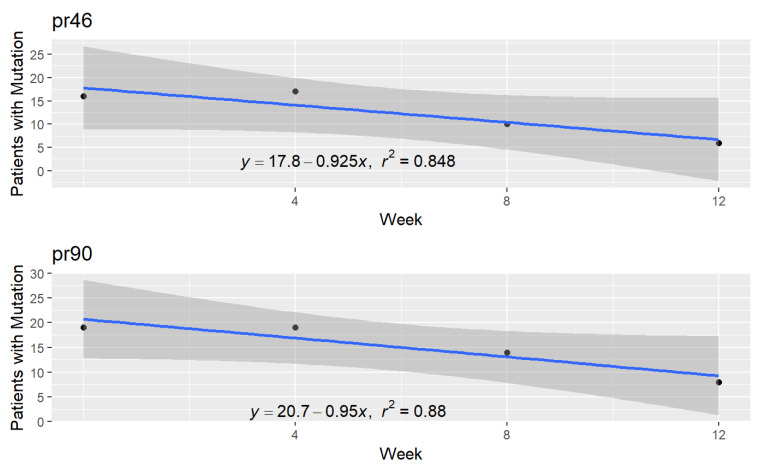
Evolution of patients with a protease inhibitor mutation at codons 46 and 90.

**Figure 5 pathogens-10-01425-f005:**
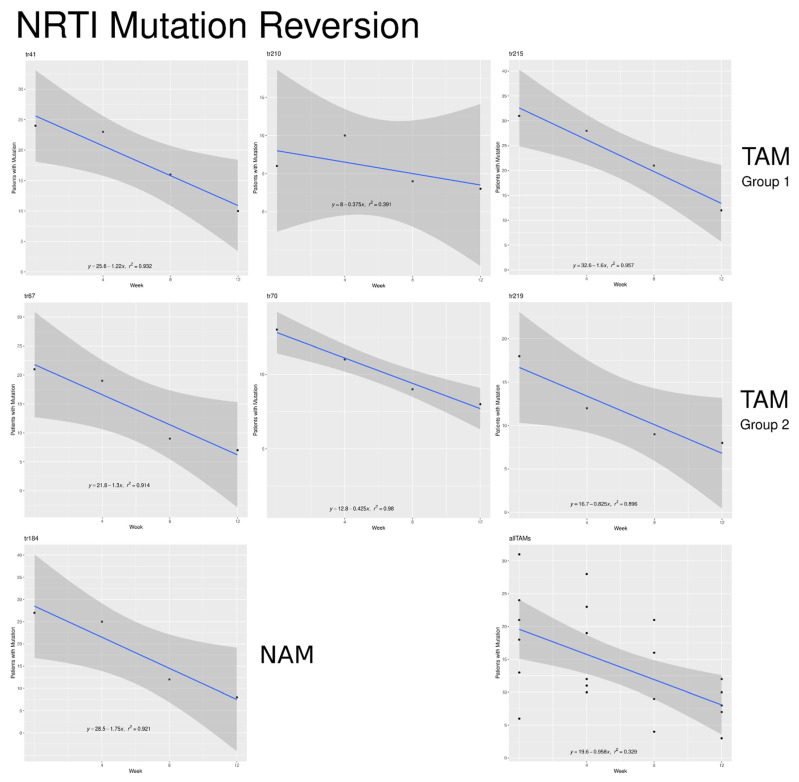
Comparison of TAM groups in the number of weeks projected to extinguish mutation. Each dot represents a single mutation within one of the three groups shown.

**Figure 6 pathogens-10-01425-f006:**
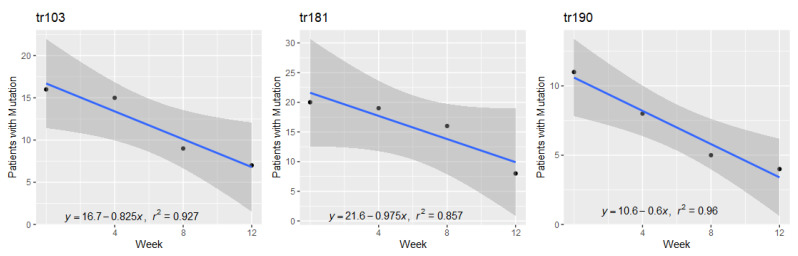
NNRTI mutation reversion.

**Figure 7 pathogens-10-01425-f007:**
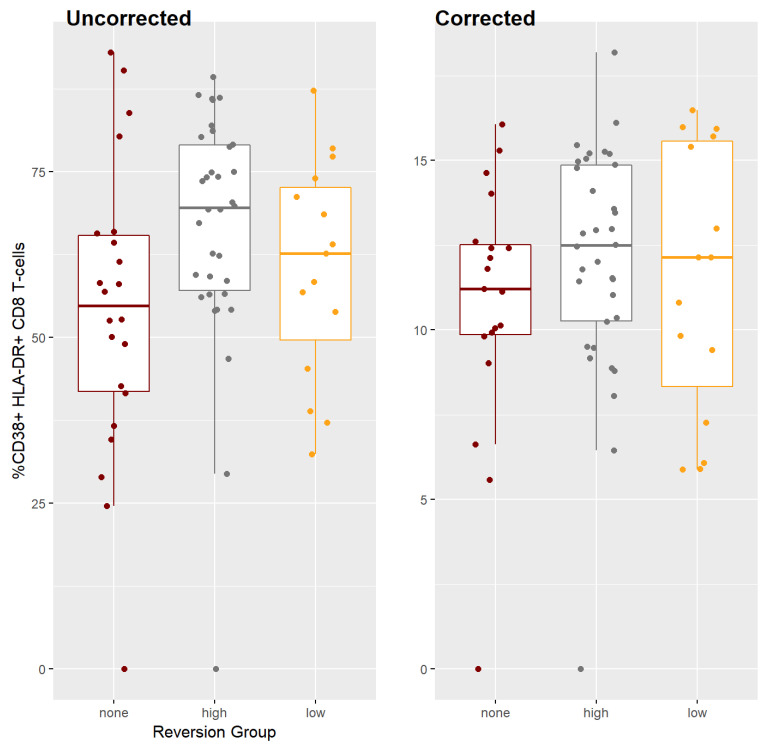
Cell activation biomarkers by reversion group.

**Figure 8 pathogens-10-01425-f008:**
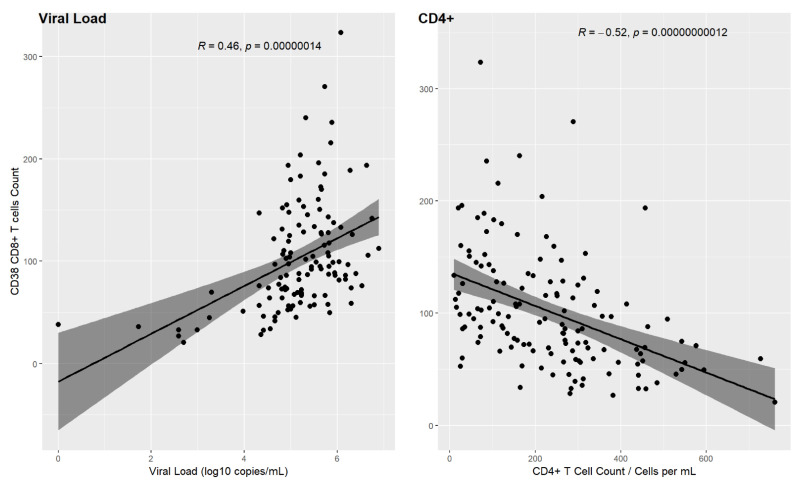
CD38+ in relation to viral load and CD4+ count.

**Table 1 pathogens-10-01425-t001:** Means (standard deviations) for viral load by reversion group (Log_10_ copies/mL).

Week	Control	All Resistant Subjects	No Reversion	Low Reversion	High Reversion
0	5.230	5.021 (0.556)	5.111 (0.730)	5.082 (0.354)	4.761 (0.532)
4	6.072	5.386 (0.511)	5.222 (0.609)	5.290 (0.614)	5.580 (0.275)
8	5.841	5.516 (0.537)	5.395 (0.610)	5.343 (0.622)	5.714 (0.368)
12	5.962	5.580 (0.597)	5.247 (0.621)	5.505 (0.513)	5.908 (0.456)
Overall Increase	0.732	0.568	0.136	0.744	0.826
*p*-value	N/A	<1.0 × 10^−6^ *	0.3021	0.0024 *	<1.0 × 10^−6^ *
n	2	36	13	15	8

* Statistically significant value (*p* < 0.05). *p*-values test whether the overall increase for each reversion category is greater than 0.

**Table 2 pathogens-10-01425-t002:** Evolution of codons in plasma sequences.

Codon	AA–Pre-ATI	AA–Post ATI	Patients with Mutation	Present in IAS List
pr10	I	L	8	Yes
pr20	R	K	5	Yes
pr20	I	K	2	Yes
pr24	I	L	2	Yes
pr33	F	L	2	Yes
pr35	D	E	2	No
pr36	I	M	3	Yes
pr46	I	M	6	Yes
pr48	V	G	3	Yes
pr54	V	I	7	Yes
pr57	K	R	2	No
pr58	E	Q	2	No
pr62	V	I	6	Yes
pr63	P	L	5	No
pr71	V	A	6	Yes
pr73	S	G	2	Yes
pr73	T	G	2	Yes
pr82	A	V	7	Yes
pr84	V	I	7	Yes
pr84	I	V	2	No
pr90	M	L	6	Yes
pr93	L	I	2	Yes
tr18	R	K	2	No
tr35	V	I	2	No
tr41	L	M	7	Yes
tr67	N	D	10	Yes
tr70	R	K	3	Yes
tr103	N	K	2	Yes
tr106	V	I	3	Yes
tr118	I	V	3	No
tr173	K	E	2	No
tr181	C	Y	5	Yes
tr184	V	M	10	Yes
tr190	A	G	2	Yes
tr210	W	L	3	Yes
tr211	K	R	3	No
tr211	Q	K	2	No
tr215	Y	T	9	Yes
tr215	F	T	2	Yes
tr218	E	D	2	No

*IAS* = IAS list of resistant related substitutions from the wild-type profile [32]. *No* = aa codon not on the IAS mutation list.

**Table 3 pathogens-10-01425-t003:** Weeks to complete reversion to wild type for PBMC mutations.

Codon	ARV Type	Weeks to 0 Patients	R^2^	Mutation Survival Probability
pr10	PI	44.000	0.143	0.184
pr20	PI	37.538	0.573	0.282
pr24	PI	17.667	0.800	0.333
pr30	PI	16.000	0.600	0.000
pr36	PI	98.500	0.246	0.483
pr46	PI	19.077	0.848	0.250
pr48	PI	16.000	0.720	0.188
pr53	PI	24.000	0.263	0.200
pr54	PI	34.261	0.354	0.423
pr63	PI	97.667	0.288	0.647
pr71	PI	33.692	0.676	0.391
pr73	PI	15.000	0.690	0.000
pr77	PI	NA	0.400	0.318
pr82	PI	22.400	0.641	0.300
pr84	PI	19.889	0.800	0.294
pr88	PI	NA	0.833	0.176
pr90	PI	21.500	0.880	0.300
tr41	NRTI–TAM 1	20.898	0.932	0.317
tr67	NRTI–TAM 2	16.357	0.914	0.217
tr70	NRTI–TAM 2	30.118	0.980	0.320
tr184	NRTI	16.000	0.916	0.214
tr210	NRTI–TAM 1	23.143	0.391	0.167
tr215	NRTI–TAM 1	19.824	0.957	0.355
tr219	NRTI–TAM 2	20.242	0.896	0.250
tr103	NNRTI	20.242	0.927	0.353
tr181	NNRTI	21.714	0.857	0.259
tr190	NNRTI	17.481	0.960	0.250

**Table 4 pathogens-10-01425-t004:** Mean weeks until complete reversion to wild type at the proviral compartment by the class antiretroviral.

Class	Mean	Standard Deviation	Number of Mutations in Study
PI	34.5	27.27	17
NRTI	20.9	4.76	7
NNRTI	19.8	2.15	3

**Table 5 pathogens-10-01425-t005:** Summary measures of cell activation by reversion group.

Reversion Group	N	Mean Cell Activation	Std Dev Cell Activation
Expression of Cell Activation Proteins			
No reversion	6	54.17	22.17
High reversion	10	66.56	17.86
Low reversion	4	60.44	16.47
Cell Activation Proteins Controlled for Log Viral Load			
No reversion	6	10.78	3.74
High reversion	10	12.07	3.41
Low reversion	4	11.466	3.95

Cell activation measured by percentage of CD8+ T-cells with both CD38 and HLA-DR present.

**Table 6 pathogens-10-01425-t006:** Differences in cell activation at baseline to week 12: no reversion and high reversion groups.

Measure	No Reversion	High Reversion	*p*-Value of Difference
cd8/cd38/hla-dr	21.637	13.252	0.3272
Corrected for Viral Load	4.916	0.658	0.1797

Test of differences based on Kruskal–Wallis rank-sum test.

## Data Availability

We have deposited all data and analyses at https://github.com/jameshunterbr/ATI accessed on 5 October 2021, with open access to the public.

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
