# Peer review of "Fitness Cost of Antiretroviral Drug Resistance Mutations on the pol Gene during Analytical Antiretroviral Treatment Interruption among Individuals Experiencing Virological Failure"

_pathogens, 2021, doi:10.3390/pathogens10111425_

Round 1

Reviewer 1 Report

In their manuscript entitled, “Fitness Cost of Antiretroviral Drug Resistance Mutations on pol Gene during Analytical Antiretroviral Treatment Interruption among Individuals Experiencing Virological Failure”, Hunter et al. analyze 36 patient samples from a 1999 Brazilian clinical study analyzing the effects of antiviral treatment interruption (ATI) on HIV-1 treatment experienced patients.  As expected with ATI, the authors find that patient viral loads increase and CD4+ counts decline.  Interestingly, they find that the time to reversion to wildype mutants was different for different classes of mutations: PI mutations> NRTI mutations >NNRTI mutations.  Knowing that different drug-mutants may persist for different lengths of time post ATI may better aid the patient in determining re-initiation of drug therapy regimens.

Overall, the manuscript is written clearly and the results reflect the author’s conclusions.  Only a few comments:

  1. Do you think there is any correlation between the half-life of PI vs NRTI vs NNRTI drugs (i.e the plasma level drug concentrations at time points post ATI) that would influence why there was a difference in the reversion rate for PI/NRTI/NNRTI mutants in the patients?
  2. If you have data on the patient subtypes, that should also be included.
  3. Please clarify for the reader what “Partial” means in Table 1 and text. Also, the patients are divided into “none”, “low” and “high” throughout the paper, but in Table 1 they are not divided that way. In the text it is stated “All resistant subjects” are the low, high & none grouped together; maybe clarify that in Table 1 in the heading for that column.
  4. Re-clarify to the reader in text line 182 and the Table 2 heading (and/or label in table) that the codons being described here are combination of protease + RT (pol). Normally the numbering would start at “1” again for RT, but here the numbering goes continuously from protease into RT, making it harder for the reader to recognize the RT mutations (ie M283V in the table is really RT M184V).

Author Response

Reviewer 1:

  1. Do you think there is any correlation between the half-life of PI vs NRTI vs NNRTI drugs (i.e the plasma level drug concentrations at time points post ATI) that would influence why there was a difference in the reversion rate for PI/NRTI/NNRTI mutants in the patients?

Reviewer 1 has raised a very important point. Therefore, we included the following text on page 16, lines 470 – 476 of the Discussion section as follows:

Compared to other classes, NNRTIs generally have a longer half-life in plasma, averaging 40 - 55 hours for efavirenz [50] and 25-30 hours for nevirapine. [51] The host's genetic profile mainly drives efavirenz's longer half life in plasma in the polymorphism of the CYP-2Bc allele. Nonetheless, the longest known half-life of NNRTIs did not affect the reversion time of mutations in PBMCs compared to other drugs from other antiretroviral classes. [52]

  1. Sustiva packge insert revised 3/2016;

https://www.accessdata.fda.gov/drugsatfda_docs/label/2016/020972s049-021360s038lbl.pdf.

  1. Vuramune package insert revised 11/2011;

https://www.accessdata.fda.gov/drugsatfda_docs/label/2011/020636s039_020933s030lbl.pdf

  1. Rodríguez-Nóvoa S, Barreiro P, Jiménez-Nácher I, Soriano V. Overview of the pharmacogenetics of HIV therapy. Pharmacogenomics J. 2006 Aug;6(4):234–45.
  2. If you have data on the patient subtypes, that should also be included.

We regret the omission from the original draft. We have included the following in the text on page 3, lines 115 – 116.

All patients had a virus of subtype B except for patients 3 and 53, who had combination subtype B/F.

  1. Please clarify for the reader what “Partial” means in Table 1 and text. Also, the patients are divided into “none”, “low” and “high” throughout the paper, but in Table 1 they are not divided that way. In the text it is stated “All resistant subjects” are the low, high & none grouped together; maybe clarify that in Table 1 in the heading for that column.

The names of the categories proved confusing for both reviewers and others. We have revised them to be clearer and to be applied consistently throughout the text and tables. The three categories are now labelled “None”, “Low” and “High”. Briefly, the three categories are based on the following logic:

  • None: an equal or greater number of mutations after the ATI than before
  • Low: fewer mutations after ATI than before, but not zero (i.e., at least one mutation remaining at Week 12).
  • High: zero mutations at Week 12 after having at least one at baseline.

We have revised the text in the Reversion Categories section (page 3, lines 118 – 133). In the text, you will also find we have clarified the meaning of the “All Resistant Subjects” column. We have also applied this definition in Table 1.

  1. Re-clarify to the reader in text line 182 and the Table 2 heading (and/or label in table) that the codons being described here are combination of protease + RT (pol). Normally the numbering would start at “1” again for RT, but here the numbering goes continuously from protease into RT, making it harder for the reader to recognize the RT mutations (ie M283V in the table is really RT M184V).

We appreciate calling our attention to this problem. We have changed Table 2 to reflect the correct codon numbers and prefaced each with either “pr” or “rt” as appropriate.

Should you have any further questions or doubts, we will be happy to respond.

Sincerely,

James R. Hunter, PhD
First Author

Ricardo Sobhie Diaz, MD, PhD
Corresponding Author

Reviewer 2 Report

Review of manuscript “Fitness Cost of Antiretroviral Drug Resistance Mutations on 2 the pol Gene during Analytical Antiretroviral Treatment 3 Interruption among Individuals Experiencing Virological 4 Failure” by James R. Hunter et al.

The study analyzed plasma and DNA HIV pol gene sequences of patients who failed anti-retroviral treatment and went through analytical treatment interruption of anti-viral treatment.  They correlate drug resistant mutations before ATI and after ATI, and correlate the groups with their viral load, CD4 and CD8 t cell counts. They conclude “Mutations in the pol gene can cause virological failure and difficulty in re-establishing effective virological suppression”

The study is worthwhile. But I find the presentation of the data and analysis are very confusing.  The authors failed to present clear definition of “groups” and often changes the name of the groups they are analyzing. For example it is not clear how they define high-reversion, low-reversion groups. The data is not presented clearly. The exact sequences of drug resistant mutations should be presented  along with the drugs the patients are being treated, and clearly labeled as drug (which drug) resistant (specific mutations) mutations versus wildtype sequences. This part is very important for the study and data analysis, and the conclusion of the manuscript.

I suggest the authors present details of drug resistant mutations of all patients before and after ATI, and include the location (HIV protein, protease, RT, ..?) and the drugs the patients receiving before ATI.  The patient viral load, CD4 counts, CD8 counts should also be listed in the table. Then the authors should justify which patients are classified as non-reversion, low-reversion and high-reversion group. The viral load, CD4, CD8 counts should include mean +- deviations of the group.

The  “PLHIV” should be defined. I could not find what it is.

Author Response

Reviewer 2:

  1. The authors failed to present clear definition of “groups” and often changes the name of the groups they are analyzing. For example it is not clear how they define high-reversion, low-reversion groups. The data is not presented clearly.

The reviewer is correct in this and, as noted above in response 3 to Reviewer 1, we have changed the titles of the categories, made their definitions clearer and applied them consistently throughout the text.  We have also reviewed the presentation of the data in that instances of lack of clarity in part stem from the confusion over categories, which we hope we have resolved. Where it also stems from inconsistency in the numbering of codons, we have tried to correct that as well. See response 4 above.

  1. The exact sequences of drug resistant mutations should be presented along with the drugs the patients are being treated, and clearly labeled as drug (which drug) resistant (specific mutations) mutations versus wild type sequences. This part is very important for the study and data analysis, and the conclusion of the manuscript. I suggest the authors present details of drug resistant mutations of all patients before and after ATI, and include the location (HIV protein, protease, RT, ..?) and the drugs the patients receiving before ATI. The patient viral load, CD4 counts, CD8 counts should also be listed in the table.

The reviewer’s recommendation is worthwhile as well in considering the reproducibility of the study. We have created two new Supplementary Tables. The first (Table S2) shows for each patient the viral loads, the CD4 count and the CD8 count recorded at each of the four sample collections. It also notes the subtypes in the cohort. This table is highlighted at page 4, lines 143 – 146. The second table (Table S3) notes whether there existed a mutation either before or after ATI for each codon studied. As well, this table lists all the antiretrovirals that a patient had taken at some point before the start of ATI. This table is mentioned in the text at page 6, lines 186 – 190.

We believe that Table 2 (pp 7 – 8) shows all the relevant mutations among the patients. The complete pol sequences themselves are quite extensive and are available both at the GitHub repository referred to in the text (https://github.com/jameshunterbr/ATI) and have been filed with GenBank.

  1. Then the authors should justify which patients are classified as non-reversion, low-reversion and high-reversion group. The viral load, CD4, CD8 counts should include mean +- deviations of the group.

We have clarified the reversion classification algorithm in Reversion Categories section (page 3, lines 118 – 133). included the standard deviations for each group in Table 1 with the means and a Supplementary Table (S2B) for the means and standard deviations of viral load, CD4 and CD8 at all 4 time points. We have not included this in the main body as it these data are not used for inference for the mutations.

  1. The “PLHIV” should be defined. I could not find what it is.

“People Living with HIV”. We have included this definition in the Abstract where the abbreviation appears (page 1, line 18).

Should you have any further questions or doubts, we will be happy to respond.

Sincerely,

James R. Hunter, PhD
First Author

Ricardo Sobhie Diaz, MD, PhD
Corresponding Author

Round 2

Reviewer 2 Report

Review of the revised “ Fitness Cost of Antiretroviral Drug Resistance Mutations on  the pol Gene during Analytical Antiretroviral Treatment  Interruption among Individuals Experiencing Virological  Failure” by Hunter et al.

The authors revised manuscript according to the reviewers comments and suggestions. Some suggested corrections for the revised manuscript are as follows:

Line 28, “was considerably less the 100% reversion 28 expected”. Should be “less than…..”

Line 166: “number of mutations” should be “drug resistant mutations” to define clearly

Lin 225: “the number of weeks it would take each mutated codon” should be “the number of weeks it would take for each mutated codon”

Line: 251: “We eliminated from further analysis codon 65” to “We eliminated codon 65 from further analysis”

Table 1  p value should indicate the p value is the result of comparison between the groups or between the group and controls. Although it has been described in the text, it should be apparent when looking at the table alone. Please check week 12 5.580(f0.597)

Need legend for Figure 1. Indicate rt: reverse transcriptase; pr: protease  

Proper Figure legend should be added to each Figure.

Lin 136, “However, viral load of the no reversion group rose until  Week 8 and then declined”. Interesting, why? Should be discussed in Discussion

Line 167, “all the divisions of blood fraction and genomic region.” The authors analyzed plasma and PBMCs, should clearly state what they have analyzed here.

The section: Protease Inhibitor (PI) Mutations at DNA level

The author’s generalizing conclusion “As seen in the table, those codons with smaller mutation survival probabilities also had generally shorter projected periods to 0 patients carrying the mutation.” is questionable.  Even for mutations with similar R2, such as Pr24 and Pr84. The authors may want to plot the mean of Mutation Survival Probability versus  the mean weeks to 0 of codon mutations with R2 above 5 to see their relationship and discuss variability and the possible cause of the variabilities between mutation survival probabilities and week to 0. 

Discussion:

“The viral load increase comes in part because an increase in active viral replication after ATI is an important factor in replenishing the proviral reservoir even for this short period, as seen for this same patient population.” Please correct the grammar of the sentence.  Also, Please break down the long sentence to make the first paragraph of the discussion more readable.

Author Response

1 October 2021

Manuscript Submission: pathogens-1266514

Dear Editor:

Thank you very much for the further reviews of our manuscript entitled Fitness Cost of Antiretroviral Drug Resistance Mutations on the pol Gene during Analytical Antiretroviral Treatment Interruption among Individuals Experiencing Virological Failure. After considering all comments and suggestions from the reviewers, we respond to all an each of the points raised by them. All modifications made in the manuscript are highlighted via the Track Change facility of the word processor.

    1. Text and Grammar Changes:

We have incorporated all the grammatical changes suggested in the Comments in the text in the locations indicated.

    1. Reviewer 2:

Correlation of the regression of time until 0 subjects showed a given mutation with the survival probability of the cohort for the same codon. We have prepared a graph that shows that the correlation is high between these two measures (0.69) and that this correlation is statistically significant using a Pearson’s Product Moment Correlation test. We suggest that the new graph appear with the Supplementary Materials (as Figure S2) as it is more methodological than virological. We have added text to the Protease Inhibitor Mutations section as follows:

We evaluated the consistency of results between the time to 0 mutations for each codon and the survival analysis. We found a correlation between these values of 0.69. This correlation was significant (p = 0.00013) using a Pearson’s Product Moment Correlation test. We have included a figure with this result in the Supplementary Materials (Figure S2).

We apologize for the delay in sending these revisions. We have suffered a perfect storm of having all our papers and research for the last year all have immediate deadlines in the last two months.

Should you have any further questions or doubts, we will be happy to respond.

Sincerely,

James R. Hunter, PhD
First Author

Ricardo Sobhie Diaz, MD, PhD
Corresponding Author